# Recombinant Measles AIK-C Vaccine Strain Expressing Influenza HA Protein

**DOI:** 10.3390/vaccines8020149

**Published:** 2020-03-27

**Authors:** Takashi Ito, Takuji Kumagai, Yoshiaki Yamaji, Akihito Sawada, Tetsuo Nakayama

**Affiliations:** 1Laboratory of Viral Infection II, Kitasato Institute for Life Sciences, Tokyo 108-8641, Japan; itot@lisci.kitasato-u.ac.jp (T.I.); di12004@st.kitasato-u.ac.jp (Y.Y.); akihito@lisci.kitasato-u.ac.jp (A.S.); 2Kumagai Pediatric Clinic, Sapporo 004-0051, Japan; tkuma@mb.infosnow.ne.jp

**Keywords:** measles AIK-C, influenza virus, hemagglutinin, recombinant virus, HI antibodies

## Abstract

Recombinant measles AIK-C vaccine expressing the hemagglutinin (HA) protein of influenza A/Sapporo/107/2013(H1N1pdm) (MVAIK/PdmHA) was constructed. Measles particle agglutination (PA) and influenza hemagglutinin inhibition (HI) antibodies were induced in cotton rats immunized with MVAIK/PdmHA. Cotton rats immunized with two doses of the HA split vaccine were used as positive controls, and higher HI antibodies were detected 3 weeks after the first dose. Following the challenge of A/California/07/2009(H1N1pdm), higher viral loads (10^7^ TCID50/g) were detected in the lung homogenates of cotton rats immunized with the empty vector (MVAIK) or control groups than those immunized with MVAIK/Pdm HA (10^3^ TCID50/g) or the group immunized with HA split vaccine (10^5^ TCID50/g). Histopathologically, destruction of the alveolar structure, swelling of broncho-epithelial cells, and thickening of the alveolar wall with infiltration of inflammatory cells and HA antigens were detected in lung tissues obtained from non-immunized rats and those immunized with the empty vector after the challenge, but not in those immunized with the HA spilt or MVAIK/PdmHA vaccine. Lower levels of IFN-α, IL-1β, and TNF-α mRNA, and higher levels of IFN-γ mRNA were found in the lung homogenates of the MVAIK/PdmHA group. Higher levels of IFN-γ mRNA were detected in spleen cell culture from the MVAIK/PdmHA group stimulated with UV-inactivated A/California/07/2009(H1N1pdm). In conclusion, the recombinant MVAIK vaccine expressing influenza HA protein induced protective immune responses in cotton rats.

## 1. Introduction

Influenza is an acute lower respiratory infectious illness caused by the influenza virus with a high mortality and morbidity in young infants and the elderly [1,2]. Unvaccinated children aged 6–35 months had an overall 11.5% infection rate of influenza each season, of which 8.7% had acute lower respiratory infection, 6.1% acute otitis media, and 1.9% pneumonia [3]. Influenza virus has three distinct subtypes, A, B, and C, and subtypes A and B infect humans causing outbreaks. Influenza virus subtype A infects migratory birds, poultry, and domestic animals, and caused the pandemics H1N1, H2N2, and H3N2, but subtype B does not. Influenza subtype B contains antigenically different Yamagata and Victoria viruses, and after the pandemic in 2009, influenza A/H1N1 Pdm, A/H3N2, B/Yamagata, and B/Victoria have been co-circulating with different distribution rates [4,5]. Seasonal influenza is a global health problem and annually updated influenza vaccines are recommended to prevent hospitalization or serious complications [6,7,8]. 

The quadrivalent split vaccine was introduced into Japan in the 2015/16 season [9]. The efficacy of the conventional split vaccine is a significant concern regarding several types of influenza vaccines against different circulating subtypes. Their efficacy was previously evaluated in double-blind randomized clinical trials [10,11], but this has become difficult to achieve because of ethical concerns regarding the disadvantage of the placebo non-immunized group. Recently, a test-negative control study was conducted, and the systematic meta-analysis revealed a vaccine efficacy of 60-70% against H1N1, 33% against H3N2, and 54% against B [12]. Vaccine effectiveness of current split vaccines was reduced to 24% against antigenically mismatched circulating H3 [12]. Thus, the effectiveness of the seasonal influenza vaccines was influenced by the cross-reactivity against matched or mismatched circulating strains [13]. In addition, the split vaccine induced poor antibody responses in young infants and did not elicit mucosal IgA and cell-mediated immunity [14]. This is because current conventional split vaccines have poor or no signal molecules to stimulate innate immunity [15,16,17].

Although the inactivated egg-derived split vaccine is the most widely used, several types of influenza vaccines have been developed; split or subunit vaccines produced by MDCK cells or the baculovirus expression system, live attenuated vaccines, and other new vaccines such as virus-like particles are under development [18]. Live attenuated influenza vaccine was developed through intra-nasal administration to compensate for the shortcomings of the split inactivated vaccine and induced nasal IgA antibody and cellular immune responses [19,20]. However, it can only be administered to subjects aged 2–49 years because of adverse events in those <2 years of age and lower effectiveness in those >50 years [7]. Influenza vaccine is recommended for all individuals >6 months of age in the US, EU, and also in Japan, but the split vaccine is less effective in young infants because its efficacy depends on the immunological CD4 memory [21,22]. Induction of the priming immune response in naïve subjects is an important issue. Live influenza vaccines are purposive tools, inducing CD4 and CD8 responses similar to natural infection [20]. In this sense, several vector viruses, expressing influenza viral protein(s), have been developed such as adenovirus [23], parainfluenza virus [24], poxvirus [25], and Newcastle disease virus [26], but all have yet to undergo preclinical or early clinical trial.

A live measles AIK-C vaccine vector platform was developed, enabling the construction of recombinant viruses expressing respiratory syncytial virus (RSV) fusion (F), G and nucleo (N) proteins, or prM and E proteins of Japanese encephalitis virus [27,28,29]. They induced protective neutralizing antibodies and cellular immune responses. The measles AIK-C vectored vaccine replicated and induced humoral and cellular immune responses in cotton rats. Cotton rats are unique experimental animals susceptible to measles virus, influenza virus, and RSV [30]. Following this study, the influenza hemagglutinin (HA) gene of circulating influenza virus H1N1 was inserted at the P/M junction of the measles AIK-C vector platform, and the protective immunity of recombinant viruses was assessed in cotton rats.

## 2. Materials and Methods

### 2.1. Virus Strains and Cells

Influenza viruses were isolated from nasopharyngeal swabs obtained from patients suspected of having influenza in Sapporo, using MDCK cells in 5% CO2 at 34 °C. Wild-type isolates of A/Sapporo/107/2013(H1N1pdm) and the reference strain of A/California/07/2009(H1N1pdm) were propagated in MEM with acetyl trypsin without adding fetal calf serum.

MDCK, 293T, and Vero cells were maintained using MEM supplemented with 10% FBS, as well as B95a cells using RPMI 1640 supplemented with 10% FBS and appropriate antibiotics.

### 2.2. Construction of the Recombinant AIK-C Vaccine Strain

The HA protein gene of A/Sapporo 107/2013(H1N1pdm) was cloned using PCR primer sets; for H1Pdm HA cloning, the primer set of Pdm ATG (5′-CCATGGAGGCAATACTAGTAGT) and Pdm TAA (5′GCGGCCGCTTAAATACATATTCTACACTG) attached with the underlined sequences of *Nco* I and *Not* I restriction enzymes was used. The cloned influenza HA genome region was inserted at the genome position 3433 of the P/M junction where an *Asc* I site was artificially introduced. Infectious recombinant cDNA was constructed and the infectious chimeric virus strain of MVAIK/PdmHA was recovered from Vero cells co-cultured with 293 T cells transfected with full-length infectious cDNA together with the helper expression plasmids of AIK-C measles N, P, and L proteins [27,28].

Vero cells were infected with MVAIK/PdmHA and culture supernatants were obtained after 1, 3, 5, and 7 days of culture. Infectivity of MVAIK/PdmHA was investigated in Vero cells to examine virus growth at different temperatures: at 33, 37, and 39 °C.

### 2.3. Experimental Protocol

The experimental protocol is shown in Figure 1. Inbred cotton rats aged 7–8 weeks were immunized intramuscularly with two doses of MVAIK/PdmHA containing 10^5^ TCID_50_/0.5 mL before and 8 weeks after the first dose, and intranasally challenged with 10^6^ TCID_50_ of A/California/07/2009(H1N1pdm) in the 13th week. For the HA split vaccine group, cotton rats were immunized with two doses of 0.1 mL of the seasonal trivalent influenza split vaccine of the 2014/15 season (Kitasato-Daiichi-Sankyo Vaccine, Tokyo, Japan) before and 3 weeks after the first dose, and were challenged with A/California/07/2009(H1N1pdm) in the 8th week. The 2014/15 seasonal influenza vaccine contained A/California/07/2009(H1N1pdm), A/New York/39/2012(H3N2), and B/Massachusetts/2/2012. Blood was sampled prior to dosing and at 1, 3, 5, and 8 weeks post-dosage.

Cotton rats were immunized with MVAIK (empty vector virus) containing 10^5^ TCID_50_/0.5 mL in a similar schedule to the MVAIK/PdmHA group. Blood was sampled prior to dosing and at 1, 3, 5, 8, 10, and 13 weeks post-dosage. The other test set was the non-immunized group (infection group), which was challenged with A/California/07/2009. Blood samples, lung tissues, and the spleen were obtained three days after the challenge in the infection, HA split vaccine, MVAIK/PdmHA, and MVAIK groups. Normal cotton rats were allocated for those non-immunized without virus challenge. Three cotton rats were used for each group. 

Experimental protocol was approved by the Committee of Experimental Animal Study of Kitasato Institute for Life Sciences (No. 16-043 and 17-021).

### 2.4. Serology

Measles particle agglutination (PA) antibody was measured using a PA antibody detection kit (Serodia^®^ Measles, FujiRebio, Tokyo, Japan). Serum samples were inactivated for 30 min at 56 °C and serial two-fold dilutions starting at 1:10 were applied for a serological assay against measles virus in duplicate. Cut-off level was <10 for measles PA.

As for the antibodies against influenza viruses, inactivated serum was mixed with receptor destroying enzyme (RDE, Denka Seiken, Tokyo, Japan) and turkey RBC to adsorb non-specific inhibitors, and sera were diluted at 1:10. Hemagglutination inhibition (HI) antibodies were examined following immunization and serial two-fold dilutions were mixed with an equal volume of 4 units of HA antigens of A/California/07/2009(H1N1pdm), A/New York/39/2012(H3N2), and B/Massachusetts/2/2012 (Denka Seiken, Tokyo, Japan). After the addition of 0.5% turkey RBC, HI titers were expressed as the highest dilutions inhibiting hemagglutination [31]. The cut-off level was <10 for influenza HI antibodies and <10 was defined as 5 for GMT calculation.

### 2.5. Recovery of Influenza Virus and Titration of Infectivity

Lung tissue was obtained three days after the challenge and homogenized. Infectivity was assayed by the appearance of CPE after placing 0.1 mL of serial 10-fold dilutions of lung homogenate on a monolayer of MDCK cells in triplicate wells in a 96 well plate. Infectivity titers are expressed as 50% TCID50/tissue 0.1 mg.

### 2.6. Immunostaining

Vero cells were infected with MVAIK/PdmHA, fixed with 1% glutaraldehyde, and stained with a 1:500 dilution of monoclonal antibody against measles N protein (Ceminon, Temecula, CA, USA) and then stained with goat anti-mouse IgG conjugated with Alex Flour 488 (ab150113, abcam, Cambridge, UK). B95a cells were infected with MVAIK/PdmHA or empty MVAIK vector virus and stained with monoclonal antibodies against measles HA protein and goat anti-mouse IgG conjugated with Alexa Flour 488 (ab150113, abcam, Cambridge, UK).

To investigate the expression of influenza HA protein in Vero and B95a cells, monoclonal antibody against influenza A/H1N1 HA protein (ab66189, abcam, Cambridge, UK) and secondary antibody against mouse IgG conjugated with Alexa Flour 568 (ab175473, abcam, Cambridge, UK) were used for influenza H1N1 HA staining.

### 2.7. Electron Microscopy

Infected Vero cells were analyzed using a scanning electron-microscope (SEM), monoclonal antibody against influenza A/H1 HA protein (ab66189, abcam, Cambridge, UK), and secondary antibody against mouse IgG conjugated with 50 nm Au colloid (Cytodiagnostics, Toronto, Canada). Samples were fixed, dehydrated, dried, and coated with 2-nm-thick platinum–palladium using a JFC-1200 Fine Coater (JEOL, Tokyo, Japan) with an accelerating voltage of 13 kV.

### 2.8. Histopathology

Lung tissues were fixed with formalin, embedded in paraffin, sectioned, and stained with hematoxylin-eosin (HE). Inflammatory responses were evaluated by scoring the microscopic findings in 20 visual fields of 12 different sections for each group, using the following criteria regarding the bronchus or bronchial area: 2 for mucus occlusion in the bronchial space, 1 for swelling of alveolar–epithelial cells, infiltration of inflammatory cells in the peri-bronchus, destruction of bronchial epithelial cells, or bleeding in the bronchial space. The following scoring method for the lung parenchyma was employed: 3 for destruction of the alveolar structure of >50%, 2 for that of 25–50%, and 1 for that of <25%, thickening of the alveolar wall, infiltration of inflammatory cells, or bleeding in the alveolar space [32]. Microscopic images were taken using a Life Technologies EVOS XL Core light microscope at 40× magnification.

Expression of influenza HA antigen was confirmed by staining with monoclonal antibody against influenza H1N1 pandemic HA antigen (ab66189, abcam, Cambridge, UK), goat antibody against mouse IgG conjugated with HRP (ab6789, abcam,), and DAB substrates (Invitrogen, Carlsbad, CA, USA).

### 2.9. Quantification of Cytokine mRNA

Lung tissues obtained three days after the challenge were homogenized and total RNA was extracted by RNA extraction kit (QIAamp Viral RNA Mini Kit, QIAGEN, Hilden, Germany). A total of 1 ug of total RNA was reverse-transcribed using the One Step Prime Script RT-PCR Kit (TaKaRa Bio, Otsu, Japan). In-house primer sets were designed referring to the NCBI database for the detection of IL-1β, IL-2, IL-4, IL-10, IL-13, IFN-α, IFN-γ, regulated on activation, normal T cell expressed and secreted (RANTES), and TNF-α mRNA in TaqMan PCR, as previously reported. In-house primer sets used in TaqMan PCR were designed in accordance with the database registered with the NCBI [33]. For each cytokine, the relative ratio of the copy number in experimental rats was calculated to that in normal cotton rats that had not been immunized.

Spleen cells were cultured under stimulation with UV-inactivated A/California/07/2009 (H1N1pdm) for 24 hours. Total RNA was extracted, reverse-transcribed, and subjected to analysis for the detection of IFN-γ mRNA. The copy number of IFN-γ mRNA was expressed by the ratio to that of β-actin.

## 3. Results

### 3.1. Virus Growth and Expression of Influenza HA Antigen

The cloned influenza HA genome region was inserted at genome position 3433 where an *Asc* I site was artificially introduced after adding GGCGCG upstream of position 3433, as shown in Figure 2A. Recombinant measles AIK-C expressing Pdm HA (MVAIK/PdmHA) was recovered by reverse genetics methods. Virus growth of MVAIK/PdmHA was examined at different temperatures in comparison with the parental AIK-C vaccine strain. No difference was observed in virus growth of MVAIK/PdmHA and AIK-C, both showing similar temperature sensitivity. Virus growth was higher at 33 °C than at 35 °C and no growth occurred at 39 °C (Figure 2B). Live staining without fixation was performed when B95 cells were infected with MVAIK/PdmHA. Measles and influenza HA proteins were expressed on the surface of infected cells (Figure 2C). Vero cells were infected with recombinant MVAIK/PdmHA and stained with monoclonal antibodies against measles N and influenza H1Pdm HA proteins. These proteins were expressed in Vero cells (Figure 2D). Infected cells were examined by SEM using monoclonal antibodies against influenza HA and Au colloids. Au colloids were observed on the surface of infected Vero cells (Figure 2E).

### 3.2. Measles PA and Influenza HI Antibodies

Measles PA antibodies developed three weeks after the first dose of MVAIK/PdmHA and MVAIK (empty vector) and increased after the second dose (Figure 3A). HI antibodies against influenza A/California/07/2009(H1N1pdm) developed after the first dose of the HA split vaccine and five weeks after the first dose of MVAIK/PdmHA, but the booster response was not significant (Figure 3B). After the first dose of the HA split vaccine, specific HI antibodies against A/New York/39/2012(H3N2) and B/Massachusetts/2/2012 were noted, while immunization with MVAIK/PdmHA induced minimal non-specific antibody titers against A/New York/39/2012(H3N2) and B/Massachusetts/2/2012 (Figure 3C,D).

### 3.3. Protection against Influenza H1N1 Challenge

Cotton rats were immunized with two doses of the HA split or MVAIK/PdmHA vaccine. They were challenged with A/California/7/2009(H1N1pdm) and the results of HE staining are shown in Figure 4A. Serious pneumonitis was noted after challenge in cotton rats without immunization and those immunized with the empty MVAIK vector, demonstrated by mucus accumulation in the bronchus and destruction of the alveolar structure. No serious pathological finding was observed in the HA split or MVAIK/PdmHA group. Histological scoring was based on inflammation, and the results in bronchial and lung parenchymal alveolar areas are shown in Figure 4B,C. Similar inflammatory scores were examined in non-immunized cotton rats and those immunized with the empty MVAIK vector after the challenge. The inflammatory score of lung tissues from cotton rats immunized with MVAIK/PdmHA was lower than that from those immunized with the HA split vaccine.

### 3.4. Recovery of Infectious Virus and Detection of the Influenza Antigen

Cotton rats were intranasally challenged with 10^6^ TCID_50_/0.5 mL of A/California/07/2009 (H1N1pdm) and the results of the recovery of infectious virus are shown in Figure 5A. More than 10^6^ TCID_50_ of influenza virus was recovered from 0.1 g of lung tissues of cotton rats immunized with the empty MVAIK vector or non-immunized rats after the challenge. Less infectious virus was recovered from cotton rats immunized with HA split vaccine compared with that from the non-immunized control group (*p* < 0.05). Markedly less infectious virus was detected in cotton rats immunized with MVAIK/PdmHA after the challenge compared with that in the HA split vaccine group (*p* < 0.01).

The influenza virus antigen was stained by monoclonal antibody against the influenza HA antigen. Broncho-epithelial cells were swollen, and HA antigens were detected in lung tissues from non-immunized rats and those immunized with the empty vector after the challenge, but only few spots were observed in those immunized with HA split or MVAIK/PdmHA (Figure 5B).

### 3.5. Quantification of IFN-α, IL-1β, TNF-α, RANTES, and IFN-γ mRNA

Lung tissue was homogenized and total RNA was extracted. The expression of IL-1β, IL-2, IL-4, IL-13, IFN-α, IFN-γ, RANTES, and TNF-α mRNA was investigated and gene expression is shown as the ratio of the copy number to that in the control group. Expression of IFN-α, IL-1β, and TNF-α are shown in Figure 6A–C. High levels of expression were observed in the empty MVAIK vector group and the split vaccine group. Expression levels in the MVAIK/PdmHA group were lower than those in the split vaccine group but not significantly. IFN-γ mRNA expression in the lung tissues was higher in the MVAIK/PdmHA group than in the split vaccine group, but not significantly (Figure 6D). The IL-2 mRNA profile was similar to that of IFN-γ mRNA. No significant expression of IL-4 and IL-13 was observed (data not shown). Similar levels of RANTES mRNA were observed in infection, MVAIK, and MVAIK/PdmHA groups (Figure 6E).

Spleen cells were stimulated with UV-inactivated A/California/07/2009(H1N1pdm) virus antigen. Total RNA was extracted and the mRNA level of influenza virus-specific IFN-γ was examined. As shown in Figure 6E, high levels of IFN-γ mRNA were observed.

## 4. Discussion

The current influenza vaccine is the HA split vaccine, which has been used for approximately 50 years. HI antibody is commonly used to measure the humoral antibody response, and an HI antibody level ≥40 is considered protective, while some researchers suggested titers ≥160 or 320 would be required in clinical settings [34]. Several limitations of split vaccines have been reported [15,16,17], and many approaches, including live vaccines, have been considered to improve the vaccine efficacy. Live attenuated intranasal vaccines have been used for more than 10 years and immune responses and effectiveness after immunization were previously summarized [20,35]. The benefits of live attenuated vaccines are the induction of both humoral and cellular immune responses, mucosal antibodies, and the development of B-cell and T-cell immune memory in naïve subjects [36]. Another challenge for live vaccines is recombinant live virus-vectored vaccines expressing influenza protein(s). Several vector viruses have been reported, and currently available live attenuated vaccine strains are considered safe and effective because of the long history of immunization practice. Modified Vaccinia Ankara (MVA) is one of the candidate vaccine strains for the development of a live vectored-platform expressing the influenza HA protein of H5N1 virus [25]. Small pox was eradicated and the vaccinia vaccine was discontinued in 1980. Adenovirus, parainfluenza virus, and Newcastle disease virus (NDV) were reported as candidate virus vectors but these are not applied in clinical vaccine usage.

The measles AIK-C strain was developed in the 1970′s and has been used mainly in Japan, Iran, and recently in Vietnam [37,38,39], and is recommended for national immunization practices at one year and 5–6 years in Japan [40]. Immunization with the influenza split vaccine is voluntary for individuals ≥6 months of age and its effectiveness in young infants is limited because of the poor stimulation of vaccine on innate immunity [14,15]. The first dose of the measles vaccine expressing influenza HA protein is administered parenterally at appropriate timing for priming against influenza and measles viruses in naïve individuals, compensating for the limitation of the current influenza split vaccine by inducing humoral and cellular immunity. AIK-C has unique characteristics of a temperature-sensitive (ts) phenotype with small plaques. Molecularly, the Pro at position 439 of the phospho (P) protein is responsible for the ts phenotype, and Leu at position 278 of the fusion (F) protein is responsible for small plaque formation [41,42]. Viral growth of recombinant MVAIK was slightly lower than that of the AIK-C vaccine strain at 38 °C, indicating the more strict ts phenotype of recombinant MVAIK [41]. The body temperature of cotton rats is approximately 38 °C and this explains the lower PA antibody level after immunization with MVAIK/PdmHA, as shown in Figure 3A. These are related to the attenuation mechanism and AIK-C is considered to be a safe live vaccine vector. Comparative clinical trials showed that the standard potency AIK-C strain vaccine induced a stronger serological response than the other high potency measles vaccines and was expected to be used as the vaccine for infants <9 months in developing countries [43,44,45].

Live recombinant measles vaccine candidates expressing influenza H1N1 HA antigen were constructed. Measles PA and influenza HI antibodies were induced in cotton rats. The second dose of MVAIK/PdmHA enhanced measles PA antibodies but not influenza HI antibodies (Figure 3A,B). Gelatin particles coated with measles antigens were used for the measles PA antibody assay [46]. Measles antigens were prepared in virus culture and there are high amounts of the nucleo (N) protein. They are considered to detect dominantly the antibody against N protein. Expression of influenza HA protein inserted at the P/M junction may be lower than N protein because of lower level of mRNA following the gene order of paramyxovirus. Influenza virus-specific humoral and cell-mediated immune responses were observed and immunization with recombinant vaccines resulted in the reduction of inflammatory responses after the challenge.

Cotton rats are unique experimental animals susceptible to RSV, measles, and influenza viruses without adaptation. In the cotton rat model, the conventional split vaccine induced significantly higher levels of HI antibodies than the recombinant measles vaccine, but led to partial protection based on pathological findings after the challenge. Higher levels of infectious virus were recovered from the lung tissues in cotton rats immunized with HA split vaccine than MVAIK/Pdm HA group. The development of influenza antibodies is insufficient to protect against infection and cellular immune responses are required for protection against and recovery from inflammatory responses.

IFN-α, IL-1β, and TNF-α are induced after virus challenge due to innate immune responses inducing inflammation. The relative expression levels of IFN-α, IL-1β, and TNF-α mRNA were lower in the lung tissue from cotton rats immunized with MVAIK/PdmHA than from those immunized with the split vaccine, reflecting the weaker inflammatory response. RANTES mRNA was detected in the lung tissues obtained from infection, MVAIK/Pdm HA, and MVAIK groups. RANTES is a chemokine to mobilize the immune cells, monocytes, eosionophils, and neutrophils at the inflammatory sites. The level of IFN-γ mRNA was higher in lung tissues from cotton rats immunized with MVAIK/PdmHA. In the MVAIK/PdmHA group, higher levels of IFN-γ mRNA were observed in spleen cell cultures stimulated with UV-inactivated influenza H1N1 Pdm antigen, demonstrating the development of cellular immunity. Further studies are required to assess CTL responses using flow cytometry.

This measles virus-vectored vaccine has several benefits including induction of both cellular and humoral immune responses, confirmed by its long-term clinical use. In addition, this platform can be used to prepare for a pandemic threat. Recombinant vaccine candidates can be generated rapidly when HA genome sequence information is available. Measles HL and the modified virus vector platform expressed H5N1 HA protein and induced cellular and antibody responses in the cynomolgus monkey [47], but the backbone measles strain is not a vaccine strain.

## 5. Conclusions

In the present study, the recombinant live measles AIK-C vaccine strain expressing influenza HA protein was constructed and was demonstrated to induce HI antibodies and protection in immunized cotton rats against influenza infection. Our study suggests this recombinant live measles AIK-C vaccine strain can induce both humoral and cellular immune responses and thus may be used as an alternative vaccine platform for other influenza stains such as A/H3N2, A/H5N1, etc.

## Figures and Tables

**Figure 1 vaccines-08-00149-f001:**
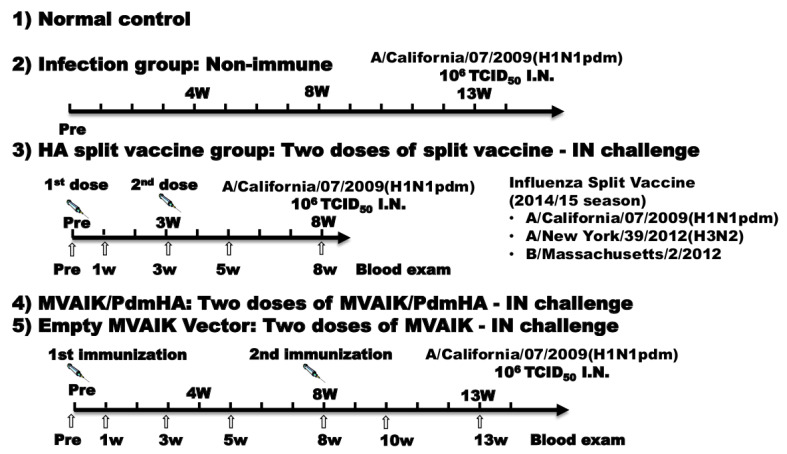
Experimental schedule. (1) Normal control group: non-immunized true control. (2) Infection group: non-immunized group was challenged with A/California/07/2009(H1N1pdm) following the same schedule as the following groups. (3) The HA split vaccine group was immunized with two doses of 0.1 mL of the seasonal HA split vaccine with a three week-interval. (4) The MVAIK/PdmHA group was immunized with two-dose recombinant virus before and 8 weeks after the first dose and challenged at 13 weeks. (5) The empty MVAIK vector group was immunized with the empty MVAIK vector. Blood was sampled prior to dosing and at 1, 3, 5, 8, 10, and 13 weeks after the first dose. Three cotton rats aged 6–8 weeks were used in each group.

**Figure 2 vaccines-08-00149-f002:**
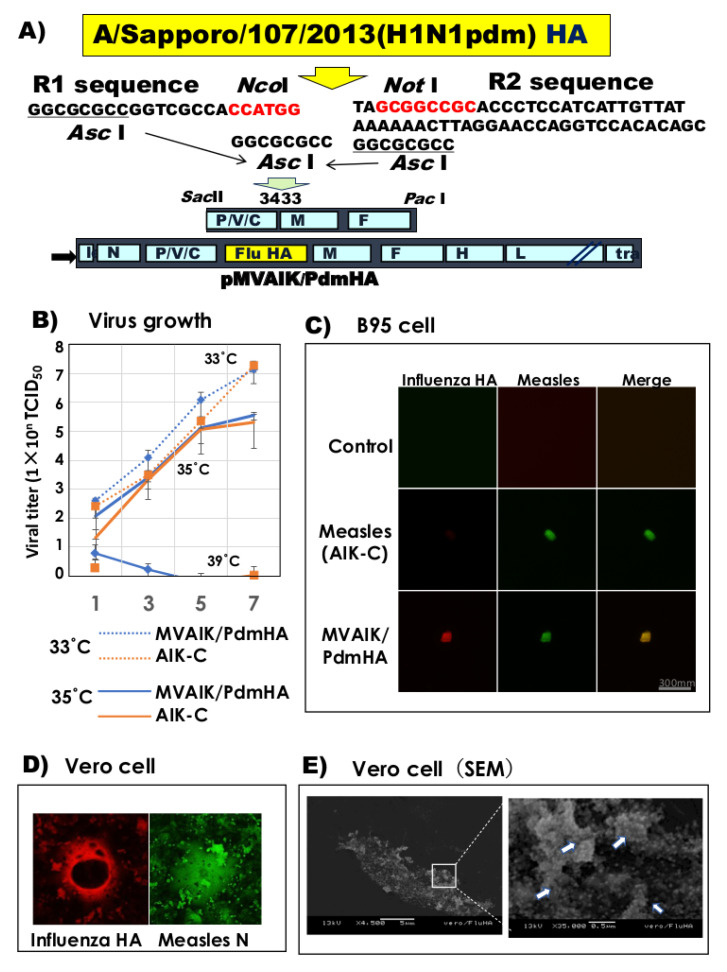
Construction of recombinant measles AIK-C expressing influenza HA protein. (**A**) The HA gene was cloned from influenza virus A/Sapporo/107/2013(H1N1pdm) HA and inserted at the P/M junction of the infectious cDNA. (**B**) Vero cells were infected with MVAIK/PdmHA at different temperatures (33 °C, 35 °C, and 39 °C) and culture fluids were obtained on days 1, 3, 5, and 7 of culture to examine the infectivity. Vertical lines represent mean ± SE. (**C**) Live staining of the cell surface expression of measles and influenza HA proteins on B95 cells. (**D**) Vero cells were infected with MVAIK/PdmHA using monoclonal antibodies against influenza HA and measles N proteins. (**E**) Electron microscope findings of the expression of influenza HA protein. Square region in the left panel is magnified, with white arrows showing the binding of Au colloids.

**Figure 3 vaccines-08-00149-f003:**
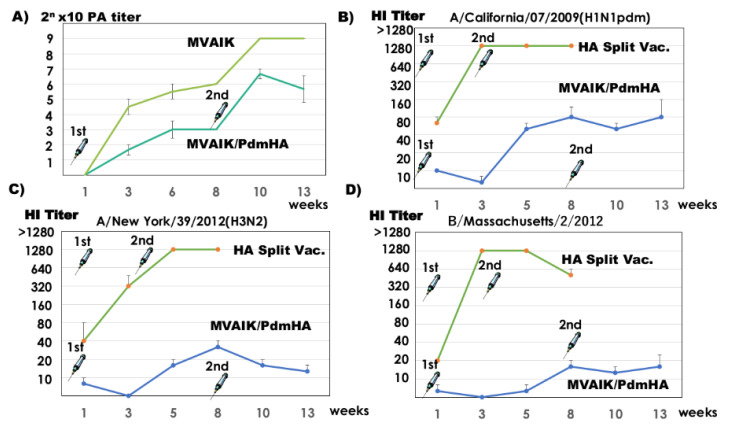
Serological responses in cotton rats immunized with influenza HA split or recombinant MVAIK/PdmHA vaccines. (**A**) Development of measles PA antibody. Vertical lines represent mean ± SE. (**B**) HI antibody against influenza A/California/07/2009(H1N1pdm), (**C**) HI antibody against influenza A/New York/39/2012(H3N2), and (**D**) HI antibody against influenza B/Massachusetts/2/2012. Vertical lines represent mean + SE.

**Figure 4 vaccines-08-00149-f004:**
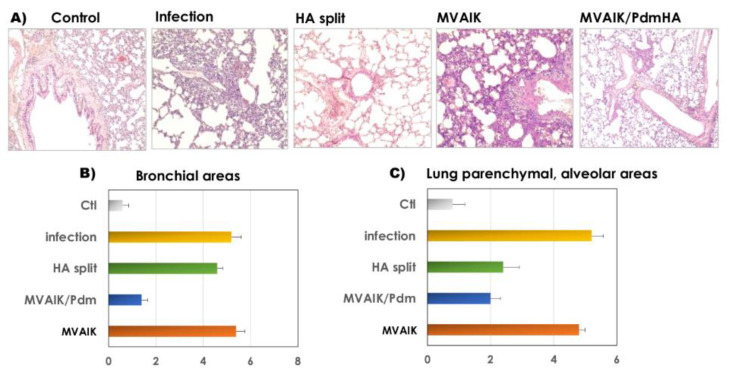
Histopathological findings of cotton rats in control, infection, HA split vaccine, MVAIK (empty vector), and MVAIK/PdmHA groups after challenge. (**A**) HE staining. (**B**) Histopathological scoring in bronchial areas: 2 for mucus and occlusion in the bronchial space, 1 for swelling of alveolar-epithelial cells, infiltration of inflammatory cells in the peri-bronchus, destruction of bronchial epithelial cells, or bleeding in the bronchial space. (**C**) Histopathological scoring in lung parenchymal and alveolar areas: 3 for destruction of the alveolar structure of >50%, 2 for that of 25–50%, and 1 for that of <25%, thickening of the alveolar wall, infiltration of inflammatory cells, or bleeding in the alveolar space. Horizontal bar represents mean + SE.

**Figure 5 vaccines-08-00149-f005:**
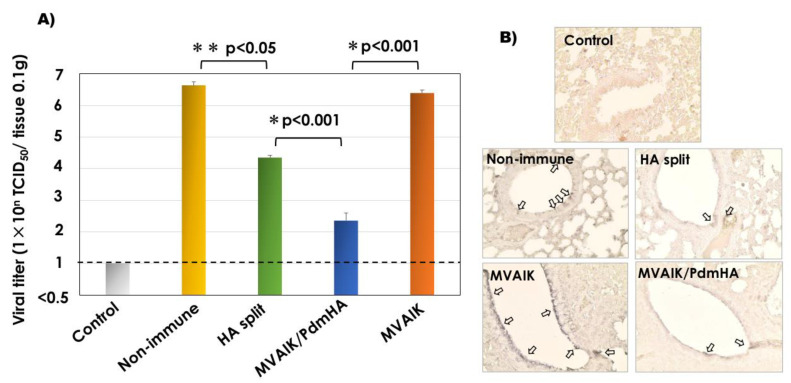
Recovery of infectious influenza virus from lung tissues after the challenge. (**A**) Infectious titers from 0.1g of lung homogenates from control, non-immune infection, HA split vaccine, MVAIK/PdmHA, and MVAIK (empty vector) groups. Horizontal dotted line shows the detection limit. Vertical column represents mean + SE. (**B**) Detection of the influenza HA antigen. Arrows show positive influenza HA staining.

**Figure 6 vaccines-08-00149-f006:**
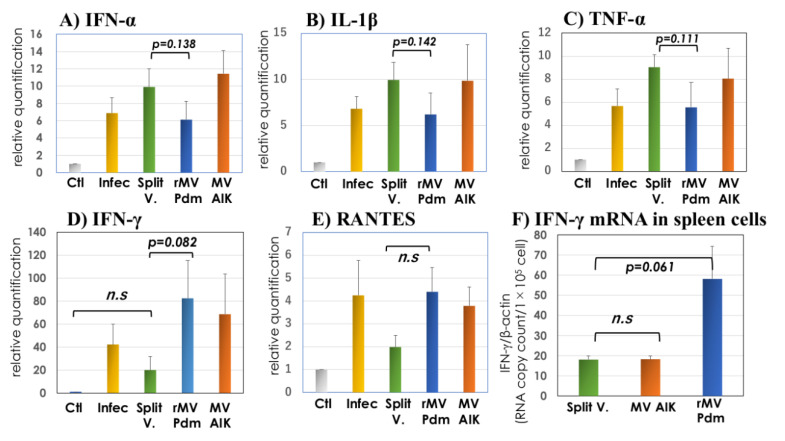
Detection of mRNA in lung homogenates and IFN-γ mRNA in spleen cell cultures stimulated with influenza antigen. Relative quantification of IFN-α (**A**), IL-1β (**B**), TNF-α (**C**), and IFN-γ (**D**), and RANTES (**E**) mRNA in lung homogenates from control (Ctl), non-immune infection (Infec), HA split vaccine (Split V.), MVAIK/PdmHA (rMVPdm), and empty vector (MVAIK) groups. (**F**) Spleen cells were cultured and stimulated with UV-inactivated A H1N1/California/07/2009 Pdm. Vertical column represents mean + SE.

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
