# Peer review of "Recombinant Measles AIK-C Vaccine Strain Expressing Influenza HA Protein"

_vaccines, 2020, doi:10.3390/vaccines8020149_

Round 1

Reviewer 1 Report

Nice and clear Abstract which continued into the introduction. Sets out appropriate literature and context of the experiments.
Expand the cloning details from line 85. I had to read this a few times and made much more sense once I read the results (from 183). Appreciate this is a balance and happy for you not to need edits but the authors may prefer to add more detail the methods. I totally get that they are immersed in the work but for someone reading like myself not embedded within the project it needed a little more explanation.

My only slight concern is the limited number of rats per group n=3. 

Data however is nice and clean. 

Overall well explained. 

Discussion valid. If anything perhaps too short but this may be in the boundaries of the journal limits. 

Author Response

Reviewer 1: Thank you for your comments and the followings are my responses.

【Comment】Expand the cloning details from line 85. I had to read this a few times and made much more sense once I read the results (from 183). Appreciate this is a balance and happy for you not to need edits but the authors may prefer to add more detail the methods. I totally get that they are immersed in the work but for someone reading like myself not embedded within the project it needed a little more explanation.                                   【Response】More detailed explanation was added in Section of Materials and Methods, line 93-102.

  1. 2. Construction of the recombinant AIK-C vaccine strain

              The HA protein gene of A/H1Pdm/Sapporo 107/2013 was cloned by PCR primer set: for H1Pdm HA cloning primer set of Pdm ATG (5’-CCATGGAGGCAATACT AGTAGT) and Pdm TAA (5’GCGGCCGCTTAAATACATATTCTACACTG) attached with the underlined sequences of Nco I and Not I restriction enzymes. Cloned influenza HA genome region was inserted at the genome position 3433 of the P/M junction where an Asc I site was artificially introduced. Infectious recombinant cDNA was constructed and the infectious chimeric virus strain of MVAIK/PdmHA was recovered from Vero cells co-cultured with 293 T cells transfected with full-length infectious cDNA together with the helper expression plasmids of AIK-C measles N, P, and L proteins [27-28].

【Comment】Slight concern is the limited number of rats per group n=3.           【Response】As the reviewer commented, number of rats is a limitation of the study. Because cotton rats are breeding in-house. They deliver 3-6 newborn babies. It would be difficult to use more than 3 rats in each group

Reviewer 2 Report

Please find below the comments for the manuscript titled “Recombinant Measles AIK-C Vaccine Strain 3 Expressing Influenza HA Protein”

  1. The authors should explain what do they mean by Pdm in their strains?
  2. The experimental protocol was confusing. The authors may have to rewrite clearly.
  3. The two control groups are not clear what do they mean by “true control without immunization and non immunized infection group 2”.
  4. The authors have to explain in detail how they cultured the influenza virus using MDCK cells by addressing the key details like culture condition, days, etc.
  5. In line number 173-174, 180 the authors should mention how or what kit they used for RNA extraction?
  6. The authors should explain on what basis these particular panel of cytokines IL-1β, IL-2, IL-4, IL-10, IL-13, IFN-α, IFN-γ, RANTES, and TNF-α were selected? There was no information on assay sensitivity, positive and negative controls.
  7. The figure 2C is very poor which doesn’t express any information needs to be improved or show appropriately.
  8. Where is the data for IL-2, IL-4, IL-10, IL-13 and RANTES?
  9. The conclusion should not be generic it needs to be more informative and addressed appropriately.
  10. Overall the language needs to be improved throughout the manuscript.

Author Response

Reviewer 2. Thank you for your comments

【Comment 1】The authors should explain what do they mean by Pdm in their strains?

【Response】A/ Sapporo 107/2013 H1Pdm was a wrong expression. Influenza virus isolate is A/H1Pdm/Sapporo/107/2013, antigenically related to A/California/7/ 2009(H1N1) pdm09 isolated in 2009 pandemic.

【Comments 2 and 3】The experimental protocol was confusing. The authors may have to rewrite clearly. The two control groups are not clear what do they mean by “true control without immunization and non-immunized infection group 2”.                 【Response】The experimental protocol was modified and following, line 107-123.

  1. 3. Experimental protocol

The experimental protocol is shown in Fig. 1. Inbred cotton rats aged 7-8 weeks were immunized intramuscularly with two doses of MVAIK/PdmHA containing 105 TCID50/0.5 mL at pre and 8 weeks after the first dose, and intranasally challenged with 106 TCID50 of A/ California/7/2009 in the 13th week. As the HA split vaccine group, cotton rats were immunized with two doses of 0.1 mL of the seasonal trivalent influenza split vaccine of the 2014/15 season (Kitasato-Daiichi-Sankyo Vaccine, Tokyo, Japan) at pre and 3 weeks after the first dose, and were challenged with A/California/7/2009 in the 8th week. The 2014/15 seasonal influenza vaccine contained A/California/7/2009(H1N1) pdm09, A/New York/39/2012(H3N2), and B/Massachusetts/2/2012. Blood samples were obtained at pre, 1, 3, 5, and 8 weeks.

Cotton rats were immunized with MVAIK (empty vector virus) containing 105 TCID50/0.5 mL in a similar schedule to the MVAIK/PdmHA group. Blood was sampled at pre, 1, 3, 5, 8, 10, and 13 weeks. The other was the non-immunized group (infection group), which was challenged with A/California/7/2009. Blood samples, lung tissues, and the spleen were obtained 3 days after the challenge in the infection, HA split vaccine, MVAIK/PdmHA, and MVAIK groups. Normal cotton rats were allocated for those non-immunized without virus challenge. Three cotton rats were used for each group.

【Comment 4】The authors have to explain in detail how they cultured the influenza virus using MDCK cells by addressing the key details like culture condition, days, etc.【Response】It was mentioned, line 86-89

Influenza viruses were isolated from nasopharyngeal swabs obtained from patients suspected of having influenza in Sapporo, using MDCK cells in 5% CO2 at 34ºC. Wild-type isolate of A/H1Pdm/Sapporo/ 107/2013 and the reference strain of A/California/7/2009 were propagated in MEM with acetyl trypsin without adding fetal calf serum.

【Comment 5】In line number 173-174, 180 the authors should mention how or what kit they used for RNA extraction?

【Response】I added the following sentence, line 191-192.

Lung tissues obtained 3 days after the challenge were homogenized and total RNA was extracted by RNA extraction kit (QIAamp Viral RNA Mini Kit, QIAGEN, Hilden, Germany).

【Comment 6】The authors should explain on what basis these particular panel of cytokines IL-1β, IL-2, IL-4, IL-10, IL-13, IFN-α, IFN-γ, RANTES, and TNF-α were selected? There was no information on assay sensitivity, positive and negative controls. 【Response】I selected IL-1β, IFN-α, and TNF-α as inflammatory cytokine, RANTES as chemokine, Il-2 and IFN-γ as Th1 cytokine. Positive controls are expression plasmids constructed by the PCR products using + and – sense primers used for the respective TaqMan real-time PCR sets. Copy numbers are calculated and for each cytokine, the relative ratio of the copy number in experimental rats was calculated to that in normal cotton rats that had not been immunized. It was mentioned line 192-203.

A total of 1 ug of total RNA was reverse-transcribed using the One Step Prime Script RT-PCR Kit (TaKaRa Bio, Otsu, Japan). In-house primer sets were designed referring to the NCBI database for the detection of IL-1β, IL-2, IL-4, IL-13, IFN-α, IFN-γ, regulated on activation, normal T cell expressed and secreted (RANTES), and TNF-α mRNA in TaqMan PCR, as previously reported. In-house primer sets used in TaqMan PCR were designed in accordance with the database registered with the NCBI [33]. For each cytokine, the relative ratio of the copy number in experimental rats was calculated to that in normal cotton rats that had not been immunized.

              Spleen cells were cultured under stimulation with UV-inactivated A/H1/California/7/2009 for 24 hours. Total RNA was extracted, reverse-transcribed, and subjected to analysis for the detection of IFN-γ mRNA. The copy number of IFN-γ mRNA was expressed by the ratio to that of β-actin.

【Comment 7】The figure 2C is very poor which doesn’t express any information needs to be improved or show appropriately.

【Response】The figure 2C presents the expression of influenza and measles HA antigen on the surface of B95a cells. B95a cells infected with MVAIK/PdmHA were stained without fixation. It means that inserted influenza HA protein is expressed on the surface of infected cells.

【Comment 8】Where is the data for IL-2, IL-4, IL-10, IL-13 and RANTES?【Response】)L-10 was not measured. IL-4 and IL-13 mRNAs were not detected. IL-2 mRNA profile was similar to that of IFN-γ. The RANTES mRNA profile was added and mentioned as following, line 290-300.

Lung tissue was homogenized and total RNA was extracted. The expression of IL-1β, IL-2, IL-4, IL-13, IFN-α, IFN-γ, RANTES, and TNF-α mRNA was investigated and gene expression is shown as the ratio of the copy number to that in the control group. Expression of IFN-α, IL-1β, and TNF-α was shown in Fig 6A, 6B, 6C. High levels of expression was observed in the empty MVAIK vector group and the split vaccine group. Expression levels in the MVAIK/PdmHA group were lower than those in the split vaccine group but not significantly. IFN-γ mRNA expressiom in the lung tissues was higher in the MVAIK/PdmHA group than in the split vaccine group, but not significantly (Fig. 6D). IL-2 mRNA profile was similat to that of IFN-γ mRNA. No significant expresssion of IL-4 and IL-13 was observed (data not shown). Similar levels of RANTES mRNA was observed in infection, MVAIK and MVAIK/PdmHA groups (Fig 6E).

【Comment 9】The conclusion should not be generic it needs to be more informative and addressed appropriately.

【Response】The conclusion is changed as following, line 386-392.

  1. Conclusions

              In the present study, the recombinant live measles AIK-C vaccine strain expressing influenza HA protein was constructed. It induced HI antibodies in cotton rats and protected influenza infection after the challenge. In comparison with HA split vaccine, the recombinant vaccine induced more efficient protective immune responses. This recombinant vectored vaccine is available for other A/H3N2, A/H5N1, A/H7N9, or B strains. This platform may improve vaccine development by inducing both humoral and cellular immune responses.

Reviewer 3 Report

Ito et al. constructed recombinant measles AIK-C vaccine expressing HA of the influenza virus A/H1Pdm/Sapporo/107/2013 strain, and the vaccine induced protective immune responses against the influenza virus in cotton rat challenge model. The study for effective live vaccine-vectored platform is very important, because current influenza HA vaccine is not completely effective in preventing influenza. However, there are two concerns to be addressed.Ito et al. constructed recombinant measles AIK-C vaccine expressing HA of the influenza virus A/H1Pdm/Sapporo/107/2013 strain, and the vaccine induced protective immune responses against the influenza virus in cotton rat challenge model. The study for effective live vaccine-vectored platform is very important, because current influenza HA vaccine is not completely effective in preventing influenza. However, there are two concerns to be addressed.

1. In this study, AIK-CMVAIK / PdmHA vaccination induced lesser anti-influenza antibody titer than HA vaccine vaccination. However, AIK-CMVAIK / PdmHA vaccination induced greater anti influenza protective response than HA vaccination. The authors indicated that higher levels of IFN-γ mRNA were observed in spleen cell cultures stimulated with UV-inactivated influenza H1N1 Pdm antigen, demonstrating the development of cellular immunity. However, the data alone is too weak to provide a direct basis for induction of a cellular immune response. Although the authors understand that direct influenza specific cellular immune responses (i.e. CTL assay for the influenza infected target cells, and influenza specific macrophage responses etc.), is there any epitope in the HA to induce HA-specific cellular immune response? To show that directly, at least, the CTL assay should be determined.

2. The vaccine used in this study is an experimental system using a polyvalent live measles vaccine expressing influenza HA. There are several reports about effects of polivalent live vaccines in which vaccine antigens of other viruses are expressed in such live vaccines (i.e. polio vaccine and varicella vaccine). Why did the authors select a measles AIK-C vaccine to produce a polyvalent live vaccine that expresses influenza HA? Consideration of the necessity or the validity of the combination seems to emphasize the importance of this paper.

Author Response

Reviewer 3. Thank you for your comments.

【Comment 1】In this study, AIK-CMVAIK / PdmHA vaccination induced lesser anti-influenza antibody titer than HA vaccine vaccination. However, AIK-CMVAIK / PdmHA vaccination induced greater anti influenza protective response than HA vaccination. The authors indicated that higher levels of IFN-γ mRNA were observed in spleen cell cultures stimulated with UV-inactivated influenza H1N1 Pdm antigen, demonstrating the development of cellular immunity. However, the data alone is too weak to provide a direct basis for induction of a cellular immune response. Although the authors understand that direct influenza specific cellular immune responses (i.e. CTL assay for the influenza infected target cells, and influenza specific macrophage responses etc.), is there any epitope in the HA to induce HA-specific cellular immune response? To show that directly, at least, the CTL assay should be determined.【Response】As the reviewer commented that production of IFN-γ mRNA alone by spleen cells stimulated with UV-inactivated A H1N1/California/07/2009 Pdm is weak to provide the cellular immunity. I agree the comments and the following sentence is added in the text. Further studies are required to assess CTL responses using flow cytometry (line 377). CTL assay was performed in human lymphocytes stimulated with UV-inactivated A H1N1/California/07/2009 Pdm or HA peptide HA 246-255. This HA peptide was used for flow cytometry in cotton rat spleen cells but failed. I think the antigenic epitope of HA protein for human might be different from cotton rats,

【Comment 2】The vaccine used in this study is an experimental system using a polyvalent live measles vaccine expressing influenza HA. There are several reports about effects of polyvalent live vaccines in which vaccine antigens of other viruses are expressed in such live vaccines (i.e. polio vaccine and varicella vaccine). Why did the authors select a measles AIK-C vaccine to produce a polyvalent live vaccine that expresses influenza HA? Consideration of the necessity or the validity of the combination seems to emphasize the importance of this paper.

【Response】AIK-C has unique characteristics of a temperature-sensitive (ts) phenotype with small plaques. Molecularly, the Pro at position 439 of the phospho (P) protein is responsible for the ts phenotype, and Leu at position 278 of the fusion (F) protein is responsible for small plaque formation [41, 42]. Viral growth of recombinant MVAIK was slightly lower than that of the AIK-C vaccine strain at 38ºC, indicating the more strict ts phenotype of recombinant MVAIK [41]. The body temperature of cotton rats is approximately 38ºC and this explaines the lower PA antibody level after immunization with MVAIK/PdmHA, as shown in Fig. 3A. These are related to the attenuation mechanism and AIK-C is considered to be a safe live vaccine vector. These were mentioned in the original manuscript.

In addition to the basic research, we added three representative references of comparative clinical trials as following, line 346-348.

Comparative clinical trials showed that the standard potency AIK-C strain vaccine induced a stronger serological response than other high potency measles vaccines and was expected to be used as the vaccine for infants <9 months in developing countries [43, 44, 45].

43) Tidjani, O.; Guérin, N.; Lecam, N.; Grunitsky, B.; Lévy-Bruhl, D.; Xuereff, C.; Tatagan, K. Serological effects of Edmonston-Zagreb, Schwarz, and AIK-C measles vaccine strains given at ages 4-5 or 8-10 months. Lancet 1989, 334, 1357-1360.

44) Bolotovski, V.M.; Grabowsky, M.; Clements, C.J.; Albrecht, P.; Brenner, E.R.; Zargaryantzs, A.I.; Litvinov, S.K.; Mikheyeva, I.V. Immunization of 6 and 9 month old infants with AIK-C, Edmonston-Zagreb, Leningrad-16 and Schwarz strains of measles vaccine. Int. J. Epidemiol. 1994, 23, 1069-1077.

45) Nkrumah, F.K.; Osei-Kwasi, M.; Dunyo, S.K.; Koram, K.A.; Afari, E.A.. Comparison of AIK-C measles vaccine in infants at 6 months with Schwarz vaccine at 9 months: a randomized controlled trial in Ghana. Bull WHO 1998, 76, 353-359.